# Proteomics of Mouse Heart Ventricles Reveals Mitochondria and Metabolism as Major Targets of a Post-Infarction Short-Acting GLP1Ra-Therapy

**DOI:** 10.3390/ijms22168711

**Published:** 2021-08-13

**Authors:** Juliana de Freitas Germano, Ankush Sharma, Miroslava Stastna, Chengqun Huang, Marianne Aniag, Angie Aceves, Jennifer E. Van Eyk, Robert M. Mentzer, Honit Piplani, Allen M. Andres, Roberta A. Gottlieb

**Affiliations:** 1Cedars-Sinai Medical Center, Smidt Heart Institute, Beverly Hills, CA 90048, USA; stastna@iach.cz (M.S.); Chengqun.Huang@cshs.org (C.H.); marianneaniag@gmail.com (M.A.); acev.angie@gmail.com (A.A.); jennifer.vaneyk@cshs.org (J.E.V.E.); robert.mentzer@cshs.org (R.M.M.J.); honit.piplani@cshs.org (H.P.); allen.andres@cshs.org (A.M.A.); 2Department of Cancer Immunology, Institute for Cancer Research, Oslo University Hospital, 0310 Oslo, Norway; ankush.sharma@medisin.uio.no; 3KG Jebsen Centre for B-Cell Malignancies, Institute for Clinical Medicine, University of Oslo, 0318 Oslo, Norway; 4Institute of Analytical Chemistry of the Czech Academy of Sciences, 60200 Brno, Czech Republic

**Keywords:** proteomics, glucagon-like peptide-1 receptor agonists, DMB, early cardiac remodeling, mitochondrion, cellular respiration, metabolism

## Abstract

Cardiovascular disease is the main cause of death worldwide, making it crucial to search for new therapies to mitigate major adverse cardiac events (MACEs) after a cardiac ischemic episode. Drugs in the class of the glucagon-like peptide-1 receptor agonists (GLP1Ra) have demonstrated benefits for heart function and reduced the incidence of MACE in patients with diabetes. Previously, we demonstrated that a short-acting GLP1Ra known as DMB (2-quinoxalinamine, 6,7-dichloro-N-[1,1-dimethylethyl]-3-[methylsulfonyl]-,6,7-dichloro-2-methylsulfonyl-3-N-tert-butylaminoquinoxaline or compound 2, Sigma) also mitigates adverse postinfarction left ventricular remodeling and cardiac dysfunction in lean mice through activation of parkin-mediated mitophagy following infarction. Here, we combined proteomics with in silico analysis to characterize the range of effects of DMB in vivo throughout the course of early postinfarction remodeling. We demonstrate that the mitochondrion is a key target of DMB and mitochondrial respiration, oxidative phosphorylation and metabolic processes such as glycolysis and fatty acid beta-oxidation are the main biological processes being regulated by this compound in the heart. Moreover, the overexpression of proteins with hub properties identified by protein–protein interaction networks, such as Atp2a2, may also be important to the mechanism of action of DMB. Data are available via ProteomeXchange with identifier PXD027867.

## 1. Introduction

Cardiovascular disease is the main cause of death worldwide and demands, every year, billions of dollars for patient treatment and hospitalization. Myocardial infarction, a leading cause of heart failure (HF), can lead to premature incapacity and early retirement, and about half of patients who develop HF die within 5 years of the diagnosis [1,2]. HF treatment consists of modulators of blood pressure such as β-receptor blockers, angiotensin II receptor blockers and angiotensin-converting enzyme (ACE) inhibitors [3], but the development of adverse post-MI remodeling with progression to heart failure is still high [4]. Therefore, there is a critical unmet need to develop additional and or new therapies to mitigate the progression of HF post-MI.

Glucagon-like peptide-1 receptor agonists (GLP1Ras) are a class of antihyperglycemic drugs that have shown benefits for heart function and reduction of the incidence of major adverse cardiac events (MACEs) in patients with diabetes. Liraglutide, a long half-life GLP1Ra tested in the LEADER clinical trial, showed a decreased risk for the primary outcome such as death by cardiovascular disease and non-fatal myocardial infarction. Moreover, the risk of all-cause death was also diminished in patients under liraglutide treatment. SUSTAIN-6 and HARMONY trials also demonstrated beneficial effects of semaglutide and albiglutide, respectively, in the reduction of MACE [5,6,7]. Despite their apparent beneficial effects, the mechanism of action is not well understood and there is no data on benefits of these agents for non-diabetic patients with post-infarction HF.

The heart is a highly energy-demanding organ and, for this reason, is enriched in mitochondria. Mitophagy, a cardioprotective cellular process of removing dysfunctional mitochondria, is triggered after a cardiac ischemic event in a cellular homeostatic response to attenuate organellar damage and apoptosis [8,9]. However, the magnitude of mitophagy activation after an ischemic event is insufficient to completely prevent adverse remodeling [10]. Previous studies observed a reduction in the fibrosis, inflammatory response and cardiomyocyte cell death in rats after a post-infarction administration of liraglutide, a known GLP1Ra. However, the role of autophagy in vivo, specifically parkin-mediated mitophagy, in limiting fibrosis and cell death following the GLP1Ra administration was unclear since the studies failed to show any effect of a GLP1Ra on remodeling in rats or were, instead, performed in cells [11,12]. Moreover, we previously showed that a short-acting GLP1Ra, specifically 2-quinoxalinamine, 6,7-dichloro-N-(1,1-dimethylethyl)-3-(methylsulfonyl)-, 6,7-dichloro-2-methylsulfonyl-3-N-tert-butylaminoquinoxaline (DMB, aka compound 2 [Sigma]) mitigates adverse cardiac remodeling in mice through improvement of mitochondrial quality control due to promotion of parkin-mediated mitophagy and mitochondrial biogenesis; we showed that an intermittent administration of DMB induced a more efficient removal of dysfunctional mitochondria in vitro and in vivo [13]. This way, we demonstrated that a post-PCAL administration of DMB had effects in mitochondrial turnover in mouse hearts. However, the extent of the broad effects mediated by DMB in the mouse heart is still unknown. 

Here, we employed a mouse model of permanent coronary artery ligation (PCAL) and global proteomics to interrogate the target proteins, organelles and biological processes affected by DMB administered after PCAL. Protein expression was analyzed on different days in the first week of postinfarction remodeling. This comprehensive approach showed a central role for mitochondria and mitochondria-associated metabolic pathways, such as fatty acid oxidation, in the beneficial effects of DMB on postinfarction remodeling. Furthermore, we reveal potential targets and pathways for therapeutic intervention to mitigate adverse cardiac remodeling.

## 2. Results

### 2.1. Mitochondrial Proteins Have a Central Role in the Course of Cardiac Remodeling

Previously, we showed that mice treated with intermittent doses of DMB (10 pmoles/25 g) showed benefits after PCAL [13]. We followed the same dose and administration schedule after PCAL and mice were sacrificed 1, 3 and 7 days after the procedure (Figure 1A). To understand the course of early postinfarction remodeling in vivo, with or without DMB therapy, we compared the expression of proteins from each time point to its preceding time point intragroup (Figure 1B, vertical). In total, 1334 proteins were identified by mass spectrometry (Spreadsheet S1). The top down and upregulated proteins were identified in the heatmap for each group, separately (Appendix A). To investigate the main modulated cellular components during early cardiac remodeling in vehicle or DMB-treated mice, we performed a separate functional analysis on the ShinyGO database. The analysis revealed that a high percentage of mitochondrial proteins were differentially regulated in the regular course of adverse cardiac remodeling, both with vehicle or DMB treatment (Figure 2 and Appendix A). Changes in mitochondrial protein expression in the control group were more pronounced on days 1 and 3 post-PCAL, where the mitochondrion appeared as the top cellular component of the differentially expressed proteins (DEPs), but were also prominent on day 7 (Figure 2A). In contrast, in DMB-treated mice, the mitochondrion and mitochondrial parts were listed in the top 30 cellular components of the DEPs on day 1 but achieved the most significant changes on days 3 and 7 post-PCAL, when it appeared as the main affected cellular component, with significant *p*-values (Figure 2B). These data demonstrate the central role of mitochondria in the course of cardiac remodeling after infarction and shows that the DMB treatment mitigates the very acute changes in mitochondrial protein expression on day 1 post-PCAL that occur in the control group.

### 2.2. Oxidative Phosphorylation and Multiple Cellular Metabolic Pathways Are Regulated in the First Week of Post-Infarction Remodeling

To understand the course of early postinfarction remodeling in vivo, with or without DMB therapy, and the cellular biological processes associated to it, we employed ShinyGO Gene Ontology (GO) analysis and core analysis on IPA to both groups, separately, to identify common and specific (unique) canonical pathways and their activity patterns (Figure 1B, intragroup comparison). 

While the metabolic process of small molecules and the redox process seem to be the main affected biological processes in both conditions during the first week post-PCAL, the subclusters identified by ShinyGO seem to differ (Appendix A). The vehicle group showed changes in the metabolism of cofactors, oxoacid, monocarboxylic and organic acids throughout the first week of cardiac remodeling. On the other hand, the analysis of samples from DMB-treated mice revealed that the effects were more pronounced on the metabolism of phosphorus and phosphate-containing compounds on day 1, in parallel with the metabolism of purines, ribonucleosides, ribonucleotides, nucleotides and ribose, which was continuous and lasted until the end of the first week post-PCAL. DMB-treated mice seem to present a more pronounced response to stress and mechanisms of regulation of biological quality 24 h after PCAL, not found among the top 30 biological processes in the control group. While the effects in cellular respiration were more pronounced on day 7 post-PCAL in vehicle-treated mice, these effects were identified early on day 3 in DMB-treated mice and lasted until day 7 post-surgery.

Through core analysis on IPA we could determine *z*-scores and predict activation (positive *z*-score) or inhibition (negative *z*-score) of canonical pathways according to the protein expression patterns of each group (Figure 3A–C and Table 1). DMB-treated mice demonstrated increased expression of oxidative phosphorylation (OXPHOS) components on day 1 compared to naïve mice, with downregulation of the sirtuin and glycolysis pathways. Compared to day 1, hearts from day 3 showed less OXPHOS, fatty acid oxidation (FAO) and amino acid degradation, but enhanced the sirtuin signaling pathway. The TCA cycle was suppressed in parallel with the repression of the NAD signaling pathway. Day 7 post-PCAL presented further suppression of FAO, OXPHOS, NAD signaling pathway and the TCA cycle, accompanied by downregulation of the metabolism of ketone bodies, acetyl-CoA biosynthesis, glutaryl-CoA and amino acid degradation, but promotion of glycolysis and the sirtuin signaling pathway in the DMB-treated group. Different from DMB-treated mice, hearts from vehicle-treated mice showed downregulation of OXPHOS, the TCA cycle and the NAD signaling pathway on day 1 post-PCAL compared to naïve mice. Moreover, FAO was downregulated on day 3 compared to day 1, and OXPHOS followed subsequent inhibition, which was also predicted on day 7, while the sirtuin signaling pathway was upregulated. Changes in NAD signaling pathway were not evident on day 3 compared to day 1 in vehicle-treated mice, but were further downregulated on day 7 post-PCAL compared to day 3, accompanied by a downregulation of proteins from the TCA cycle in the vehicle-treated group. These data suggest that the timing for mitochondrial turnover is different between control vs. treated mice after PCAL. DMB’s mechanism of action may also involve sustained changes to multiple cellular metabolic pathways.

### 2.3. Cross-Comparative Analysis Identifies Mitochondrial Compartments as the Main Cellular Components of DEPs

Following the separate analysis on both groups throughout the course of infarction, we performed a cross-comparative analysis between DMB and vehicle-treated mice on days 1, 3 and 7 post-PCAL with the purpose of identifying cellular components and biological processes involved in the mechanism of action of DMB (Figure 1B, horizontal; Spreadsheet S3). On day 1 post-PCAL, we found 45 DEPs; 16 of them were significantly downregulated and 29 significantly upregulated by DMB. A total of 83 proteins were uniquely found in the control group while 6 were unique from the DMB group. Day 3 after PCAL is when most of the changes were identified in our analysis and a total of 86 proteins were found to be differentially expressed: 45 proteins were upregulated by DMB and 41 were downregulated; moreover, 25 were uniquely found in the DMB group, while 8 unique proteins were identified in the vehicle-treated group. On day 7 post-PCAL, 28 out of 47 DEPs were downregulated and 19 were upregulated; 9 were uniquely found in the vehicle-treated group and 11 were unique in the DMB-treated group (Figure 4A and Table 2). ShinyGO was used to analyze the cellular components from the DEPs on each day. Mitochondrial compartments were identified on days 1, 3 and 7 as the main cellular components (Figure 4B–D). These data demonstrate that the DMB mechanism of action may rely on its effects in mitochondria, mainly, and that its beneficial effects in the ischemic heart may be associated to changes in mitochondrial function. 

### 2.4. The Cross-Comparative Analysis Reveals a Predominant Role of DMB in Regulating the Redox Signaling, ATP Production and Overal Metabolism during Cardiac Ischemia

GO analysis on ShinyGO was used to identify biological processes from the unique proteins and DEPs between DMB vs. vehicle-treated mice on days 1, 3 and 7 post-PCAL, followed by a parallel analysis by ClueGO with the generation of networks for biological processes and identification of unique proteins and DEPs associated to the pathways. The main biological processes that were identified by ShinyGO revealed a central role of DMB in regulating small molecule metabolic processes and redox signaling throughout the first week post-PCAL (Appendix A). Day 1 (Appendix A) demonstrated an overall impact of DMB in the ATP metabolic process and in the metabolism of carbohydrate derivatives, culminating in stronger effects in the generation of precursor metabolites and energy on day 3 (Appendix A), with evident alterations in lipid metabolism on day 7 (Appendix A). A parallel analysis by ClueGO showed similar biological processes compared to the ShinyGO analysis (Figure 5 and Appendix A). Day 1 demonstrated major effects of DMB on cellular respiration and on the purine ribonucleoside triphosphate metabolic process, which includes ATP synthesis. These changes were paralleled by DMB regulation of cell death and ROS metabolism, with alterations in other metabolic pathways including ketones and glucose-6-phosphate (Figure 5A and Appendix A). On day 3, the most enriched pathways identified by ClueGO were related to the generation of precursor metabolites and energy, electron transfer activity and ATP synthase activity; ROS metabolism continued to be a DMB-targeted pathway and the first effects on FAO and FA transport were manifested (Figure 5B and Appendix A). On day 7, DMB therapy showed the effects on the aerobic electron transport chain, an enhanced impact on FAO and differential modulation of acyl-CoA and branched-chain amino acid metabolism. These changes were accompanied by the continued regulation of the metabolic process and cell redox homeostasis (Figure 5C and Appendix A). The data demonstrate that intermittent administration of DMB beginning 2 h after infarction continuously modulates ROS and multiple metabolic pathways throughout the course of early remodeling, especially FAO, and this provides novel and valuable insights into the beneficial effects of GLP1 receptor agonism in the heart.

### 2.5. Protein–Protein Interaction Networks Reveal Dynamic Changes in Protein Expression in Early Remodeling

To dive deeper into the mechanism, we generated protein–protein interaction networks from unique proteins and DEPs on each time point using Cytoscape, which can identify hub proteins central to the beneficial effects of DMB. Unique, down and upregulated proteins with PPI were labeled in black, green or red, respectively. Few proteins are differentially expressed in more than one time point (Figure 6A) and only one (CLPP) was concomitantly differentially expressed on days 1, 3 and 7 after permanent ischemia. Unique proteins and DEPs from day 3 and 7 showed more interactions with partners than unique proteins and DEPs from day 1 (Table 3). On day 1, AT2A2 (aka Serca2 or Atp2a2), a cardiac Ca2+ ATPase showing upregulation on days 1 and 3 after surgery in the DMB-treated group, was identified as a hub protein, with the largest number of interactions (Figure 6B and Table 3). Day 3 after surgery presented the largest number of unique proteins and DEPs with PPI (Figure 6C and Table 3). FXR1 and TAU (Mapt), the proteins unique to the vehicle-treated group and DMB-treated group on days 3 and 7, respectively, showed hub properties with a high number of interactions in the corresponding network for those days. FACD2 (Fancd2) was identified as a first order interacting protein with hub property (large number of interactions) in all three time points and is a crucial regulator of protein–protein interactions. Altogether, these data suggest that long lasting changes in protein expression happen to a select and small group of proteins throughout the course of early remodeling and that FACD2 may have a central role in this mechanism.

## 3. Discussion

Clinical trials have shown a reduced occurrence of MACE in diabetic patients receiving GLP1Ra therapy [5,6,7]. However, there is a lack of information regarding the mechanism of action of GLP1Ra in the heart under non-diabetic conditions. Previous data demonstrated the involvement of autophagy, specifically parkin-mediated mitophagy, in the alleviation of post-infarction inflammation and cell death in mice treated with liraglutide [11]. Moreover, our group previously reported the benefits of treatment after PCAL with a small-molecule, short-acting GLP1R agonist—DMB—to reduce adverse cardiac remodeling in lean mice, and these benefits were associated with parkin-mediated mitophagy [13]. Both studies reveal mitochondria as important to the mechanism of action of GLP1Ra; however, the consequences of GLP1 receptor agonism in the ischemic heart is not well understood, and the aim of this work was to identify the main biological processes regulated by the differentially expressed proteins following DMB treatment.

Our first approach was to evaluate changes in protein expression in whole ventricles throughout the first week of cardiac remodeling. This analysis showed that mitochondrial proteins comprised a huge percentage of unique proteins and DEPs throughout the first week after infarction and placed the mitochondrion and mitochondrial compartments as the main cellular components. It also identified key differences in cell metabolism regulation when vehicle and DMB-treated mice were analyzed individually. As part of the natural history of postinfarction remodeling [14], proteins involved in OXPHOS, mitochondrial and non-mitochondrial cell metabolism were downregulated one day after infarction in vehicle-treated mice; however, this overall downregulation of metabolism was delayed in hearts from DMB-treated mice, which presented increased OXPHOS and downregulation of glycolysis. It is possible that the acute administration of DMB 2 h after the infarction preserves a population of more resilient and stress-responsive mitochondria in the first hours, evidenced by the lack of changes in the expression of proteins associated to mitochondrial metabolism, such as fatty acid oxidation and TCA cycle, between hearts from day 1 post-infarction vs. naïve hearts. This way, it is reasonable to suggest that hearts from DMB-treated mice would show more similarities to hearts from naïve mice than hearts from vehicle-treated mice in terms of metabolism. However, this would be later followed by an overall downregulation of metabolic pathways in DMB-treated hearts, mainly mitochondrial metabolism. 

Hearts from DMB-treated mice also affected mechanisms of regulation of ROS and nitric oxide (NO) production; besides its apparent increase on the first days after PCAL, possibly driven by the upregulation of OXPHOS [15,16], it culminated with a downregulation of this pathway on day 7 post-PCAL. Several pathways such as senescence, necroptosis and death receptor signaling were downregulated under DMB therapy. Instead, besides the downregulation of NO and ROS, the vehicle group showed an active pattern in inflammatory pathways on day 7 post-PCAL, such as “Fcy Receptor-mediated Phagocytosis in Macrophages and Monocytes” and “Leukocyte Extravasation Signaling”, consistent with an influx of macrophages that may occur specially at this time point [17,18,19]; a week after surgery, hearts from vehicle-treated mice demonstrated upregulation of pathways related to necroptosis and cardiac hypertrophy, which are closely related to exacerbated inflammatory response [20,21]. Together, these data suggest the exacerbated response in the vehicle group. The results are in accordance with our previous findings, which showed that DMB suppresses inflammation in mouse hearts compared to vehicle-treated mice [13].

Our second approach compared differences in protein expression in whole ventricles of DMB vs. vehicle-treated mice at each time point. This analysis identified possible mechanisms of action of DMB in the first week post-infarction, which were mostly associated to the modulation of proteins involved in cellular respiration and metabolism, ATP synthesis and ROS generation. It is important to observe that a parallel analysis from ClueGO showed similarities to the analysis from ShinyGO, which is an updated tool for gene-set enrichment analysis. Overall, DMB targeted cellular respiration and energy generation, possibly due to the modulation of lipid metabolism that would be more noticeable on day 7. Besides the possibility that indirect mechanisms are involved in the effects of GLP1 receptor cardioprotection in patients with diabetes [22], it is also possible that direct mechanisms may be involved. Previously, we showed no effects of DMB in circulating glucose levels in lean mice [13] and our current data shows that DMB therapy modulates several proteins related to energy metabolism using the same in vivo model of PCAL. This might reflect that DMB (and possibly other GLP1Ra) modulation of energy metabolism in the heart is independent of insulin. Moreover, the DMB mechanism of action implicates continuous regulation of ROS and less reliance on glycolysis in the first hours following infarction; this might reflect the preservation of mitochondrial metabolism. The sustained regulation of ROS by DMB would be mediated through the effects of DMB on the hydrogen peroxide metabolic process mainly, but also in the glutathione metabolic process (day 1), which is the most abundant antioxidant in the heart [23]. 

Finally, the PPI network analysis identified dynamic changes in protein expression between the groups throughout the course of early remodeling. Atp2a2 (Serca2), which is overexpressed in DMB-treated mice on days 1 and 3, showed a high degree of PPI on day 1. Loss of Atp2a2 expression is well-established as a contributing factor to HF and overexpression of this protein improves cardiac metabolism, coronary blood flow and cardiac function [24,25]. Fxr1, a hub protein and unique from vehicle-treated hearts on day 3 but only overexpressed in DMB-treated hearts on day 7, is associated with the development of left ventricle dilated cardiomyopathy [26]. On the other hand, Mapt (Tau) was uniquely expressed in DMB-treated hearts on day 7 and was also identified as a central hub protein. Mapt is highly expressed in neurons and associated to Alzheimer’s disease. However, there is evidence for the presence of this protein in the cardiac tissue, which expression would be essential to improve cardiovascular performance [27]. Interestingly, Fancd2, a member of the Fanconi Anemia proteins, was the only first-degree interacting protein identified as a hub protein in all time points. Disruption in the expression of this protein might show greater effects in DMB than vehicle-treated mice. Fancd2 is required for Parkin-mediated mitophagy and can suppress inflammasome activation [28]. This analysis demonstrated that the course of early remodeling follows separate fates potentially by the induction of the expression of proteins unique to a specific group throughout the days. The maximum number of unique proteins with PPI was achieved on day 3 (combining the respective day node color to the black protein label). These unique proteins can be used as important targets for the mitigation of adverse remodeling in future studies and explain the potential detailed mechanism of action of DMB. 

We conclude that the beneficial effects of early intermittent DMB therapy after a myocardial infarction are due primarily to its effects on the inflammatory pathway and mitochondrial metabolism, mostly FAO. To our knowledge, we are the first to analyze the proteomics of post-infarction therapy with a GLP1R agonist. There is a growing number of publications and clinical studies utilizing these molecules and associating the benefits of this therapy to heart disease. We believe that studying the proteomics of DMB therapy in mouse hearts will help us better understand the underlying mechanisms involved in the beneficial effects of this class of drugs in the mitigation of adverse cardiac remodeling and in the development of alternative therapies to the treatment of this pathophysiological process. 

## 4. Materials and Methods

Animal Ethics: All the procedures followed the guidelines from the National Institute of Health and were approved by the Institutional Animal Care and Use Committee of Cedars-Sinai Medical Center under the protocol IACUC-5000. 

### 4.1. Permanent Coronary Artery Ligation (PCAL) and DMB Treatment 

A PCAL mouse model was used to assess the effect of DMB administration post infarction on protein expression, as previously described [13]. C57BL/6J mice (Jackson Laboratory, Bar Harbor, ME, USA) received injections of 50 μL of the vehicle (DMSO; cat # D4540; Millipore Sigma, Burlington, MA, USA) or 10 pmol of DMB (cat # G8048; Sigma, Saint Louis, MO, USA;) in 50 µL of DMSO 2 h after PCAL and again 2 and 5 days after PCAL. 

Heart harvesting: Mice were sacrificed 1, 3 and 7 days after PCAL and whole hearts, including non-treated hearts from naïve mice, were harvested for proteomic analysis (*n* = 3/group; 21 samples in total). Briefly, the mice were individually anesthetized with isoflurane (cat # 501017; Vet One, Boise, ID, USA) and sacrificed by cervical dislocation. The thorax was cleaned with a solution of 70% ethanol and opened, and the heart was quickly harvested and perfused with a solution of phosphate buffered saline (cat # 510224; Vet One, Boise, ID, USA). The entire left ventricle (LV), including the infarct/scar, border zone and remote region was frozen in a 1.5 mL tube in liquid nitrogen. Later, the frozen hearts were ground to a powder using a mortar and pestle in liquid nitrogen and aliquots of approximately 10 mg were stored at −80 °C for further proteomic analysis.

### 4.2. Proteomic Analysis

The ground heart tissue of each sample (3 mg) was mixed with 150 µL of lysis buffer containing 6 M urea (cat # BDH4602; BDH, Poole, United Kingdom) and 0.05 M Tris-HCl, (cat # M108; Amresco, Dallas TX, USA) pH 8, in a microtube (fluorinated ethylene propylene; cat # MTWS-MT-RK; Pressure Biosciences Inc., South Easton, MA, USA) with a corresponding microcap (cat # MC150-96; Pressure Biosciences Inc., South Easton, MA, USA) and lysed by pressure cycling technology using a barocycler (NEP 2320; Pressure Biosciences Inc., South Easton, MA, USA) for 60 cycles at 45,000 psi. After lysis, the microtube with a content was placed in a 1.5 mL Eppendorf tube and centrifuged at 18,000× *g* for 15 min at 4 °C. The supernatant was collected, an additional 150 µL of lysis buffer was added to the tissue pellet and the barocycling process was repeated for 30 cycles at 45,000 psi. After the centrifugation, both supernatants were combined in a 1.5 mL LoBind (Eppendorf, Hamburg, Germany) tube and the protein concentration was determined by the Pierce BCA protein assay kit (cat # 23225; Thermo Fisher Scientific, Waltham, MA, USA). For each sample, the volume corresponding to 150 µg of protein was mixed with a lysis buffer to a total volume of 110 µL and 330 µL of 0.05 M ammonium bicarbonate (cat # BDH9206; BDH, Poole, UK) was added to dilute the urea in the sample prior to protein reduction/alkylation and trypsin digest. A total of 30 µL of 0.2 M DTT (cat # BP172; Thermo Fisher Scientific, Waltham, MA, USA) in 0.05 M ammonium bicarbonate (final concentration: 0.012 M) was added and the solution was incubated at 37 °C for 45 min on a shaking platform. For protein alkylation, 30 µL of 0.4 M iodoacetamide (cat # RC-150; G-Biosciences, Saint Louis, MO, USA) in 0.05 M ammonium bicarbonate (final concentration: 0.024 M) was added and incubated at RT for 45 min in the dark. The proteins in the sample (pH 8) were digested using trypsin (cat # V5111; Promega, Madison, WI, USA) at a 1:50 (trypsin:protein) ratio at 37 °C overnight with shaking. The next day, the sample solution was cooled to room temperature, 10% formic acid (cat # A117, Optima LC/MS grade; Fisher Chemical, Waltham, MA, USA) was added to pH 2–3 and the sample was desalted using µ-elution HLB 96-well plate 30 µm (cat # 186001828BA; Oasis Waters, Milford, MA, USA). The sample eluates were dried in the SpeedVac concentrator (cat # SPD2010; Thermo Scientific, Waltham, MA, USA) and stored at −80 °C.

Mass spectrometry: Each sample with digested peptides was resuspended in 50 µL of 0.1% formic acid and 1 µL was injected into LC/MS/MS Ultimate 3000 nano LC connected to an Orbitrap Elite mass spectrometer (Thermo Fisher Scientific, Waltham, MA, USA) via an EasySpray ion source. First, the sample was loaded into the trap column PepMap100 C18; 300 µm i.d. × 5 mm, 5 µm, 100 Å (cat # 160454; Thermo Fisher Scientific, Waltham, MA, USA) and then separated using the PepMap RSLC C18 column 75 µm i.d. × 25 mm; 2 µm, 100 Å (cat # 164536, Thermo Fisher Scientific, Waltham, MA, USA). With a flow rate of 300 nL/min, the linear gradient 5–35% of B for 90 min was applied followed by 35–95% of B for 3 min, then 95% of B for 7 min and re-equilibration at 5% of B for 25 min at 400 nL/min. The mobile phase A was 0.1% formic acid and the mobile phase B was 0.1% formic acid in acetonitrile (cat # A996, Optima LC/MS grade; Fisher Chemical; Waltham, MA, USA). The nanosource capillary was set to 275 °C and the spray voltage was 2 kV. MS1 scans were acquired in the Orbitrap Elite at a resolution of 60,000 FWHM (400–1700 *m*/*z*) with an AGC target of 1 × 10^6^ ions over a maximum of 250 ms. MS2 spectra were obtained for the top 15 ions from each MS1 scan in the CID mode in the ion trap with 30,000 FWHM and a target setting of 1 × 10^4^ ions, with an accumulation time of 100 ms and isolation width of 2 Da. Two mass spectrometry replicates were measured for each sample (total 42 analysis).

Protein identification: To search the spectra against the UniProtKB database (UP mouse, reviewed, canonical), the Sorcerer™-SEQUEST^®^ platform (Sage-N Research, Inc., Milpitas, CA, USA) was applied as a protein search engine followed by Scaffold 4.5.3. software for post-search analysis (www.proteomesoftware.com accessed on 18, 23, 24 and 25 October 2018). The search parameters were: semi-enzyme digest using trypsin (after KR/-), up to 2 missed cleavages, carbamidomethyl C as a static modification, precursor mass range from 400 to 4500 amu, peptide mass tolerance of 50 ppm and monoisotopic fragment mass type. In Scaffold, the threshold values for protein and peptide identification probabilities were both set at 95% with a minimum number of peptides to identify the protein as 2.

### 4.3. Bioinformatic and Statistical Analysis

All proteins showing *p* < 0.05 were considered as differentially expressed (DE). Heatmaps with the top 20 DEPs were obtained using the Morpheus software from the Broad Institute (https://software.broadinstitute.org/morpheus, accessed on 18 October 2018) using log2 fold-change values. The PPI networks for all proteins showing quantification and differential quantification were constructed using PPI obtained from a recent release of BioGRID version 3.5.183 [29]. The PPI dataset constituted 52,283 interactions (experimentally validated) among 13,785 proteins. The networks were visualized using Cytoscape version 3.7.2 [30]. The centrality statistics computing topological importance of proteins at the system level in PPI were computed using NetAnalyzer [31]. The hub proteins are the highly connected entities in PPI networks and removal of a hub protein disrupts the network [32,33]. Gene Ontology (GO) classification and functional enrichment analysis were performed on unique proteins and DEPs using ShinyGO v0.61 [34] and the default mode of ClueGO version 2.5.8 [35], which provided networks for biological processes with identification of their main DE or unique proteins. The IPA toolkit was used to identify canonical pathways and predict activation/inhibition of pathways through the z-score analysis (activation = positive *z*-score; inhibition = negative *z*-score). 

## Figures and Tables

**Figure 1 ijms-22-08711-f001:**
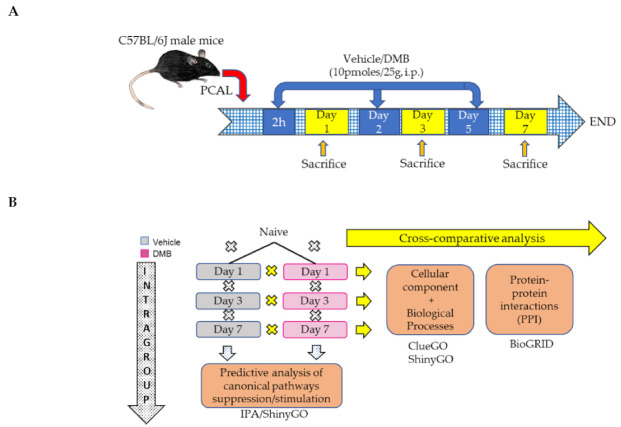
Schematics. C57BL/6J male mice (except naïve) were treated with either 10 pmoles of DMB or vehicle (DMSO) 2 h after permanent coronary artery ligation (PCAL) surgery and every other day or two (*n* = 3/group). Mice were sacrificed after 1, 3 or 7 days of surgery. (**A**) An in vivo model of PCAL surgery and treatment; (**B**) in silico analysis schematic: samples followed the intragroup analysis, when each time point was compared to its preceding time point in each group, separately; or samples followed a cross-comparative analysis, when protein expression from DMB-treated mice was compared to protein expression in the vehicle-treated group at each time point. i.p., intraperitoneal; IPA, ingenuity pathway analysis.

**Figure 2 ijms-22-08711-f002:**
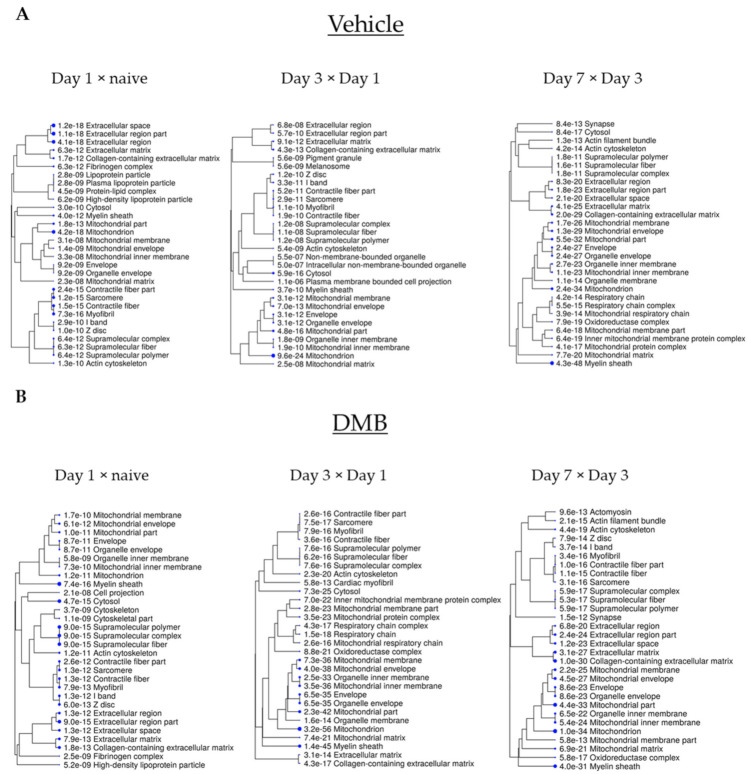
Top 30 cellular components of the unique proteins and DEPs in the intragroup comparison. The expression of proteins from each time point was compared to its preceding time point intragroup to identify unique proteins and DEPs (*p* < 0.05). A list containing all unique proteins and DEPs at each time point was uploaded on ShinyGO v0.61 and cellular components were obtained. Cellular components were clustered according to the amount of proteins they have in common, and the size of the blue dots represent *p*-values (the bigger the size, the lower the *p*-value). (**A**) Cellular components of unique proteins and DEPs in the vehicle-treated group; (**B**) cellular component of unique proteins and DEPs in the DMB-treated group.

**Figure 3 ijms-22-08711-f003:**
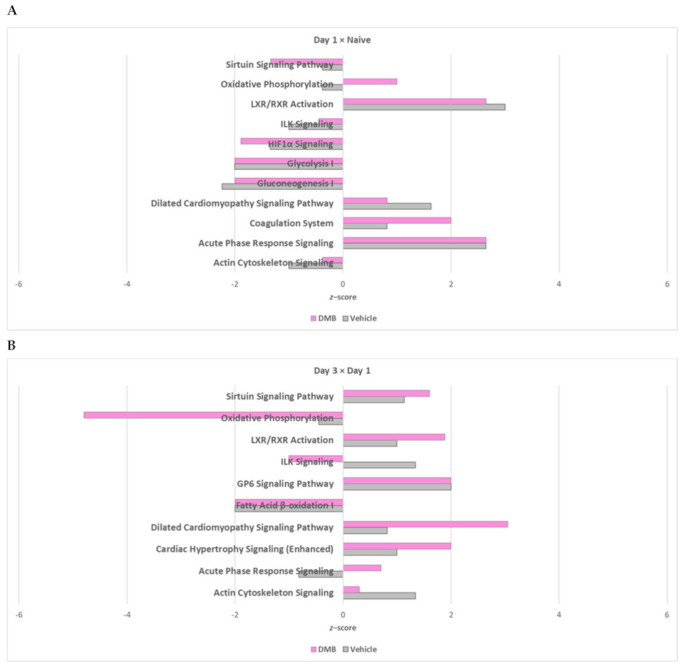
Common IPA canonical pathways with predicted *z*-scores for the course of early remodeling in each group. The expression of proteins from each time point was compared to its preceding time point intragroup. The DEPs and unique proteins identified in each comparison were uploaded on IPA for subsequent core analysis and *z*-scores were identified. *z*-scores higher than 0 predict activation and *z*-scores lower than 0 predict inhibition of a pathway. (**A**) Common canonical pathways of day 1 post-PCAL vs. naïve hearts; (**B**) common canonical pathways of day 3 post-PCAL vs. day 1 post-PCAL hearts; (**C**) common canonical pathways of day 7 post-PCAL vs. day 3 post-PCAL hearts.

**Figure 4 ijms-22-08711-f004:**
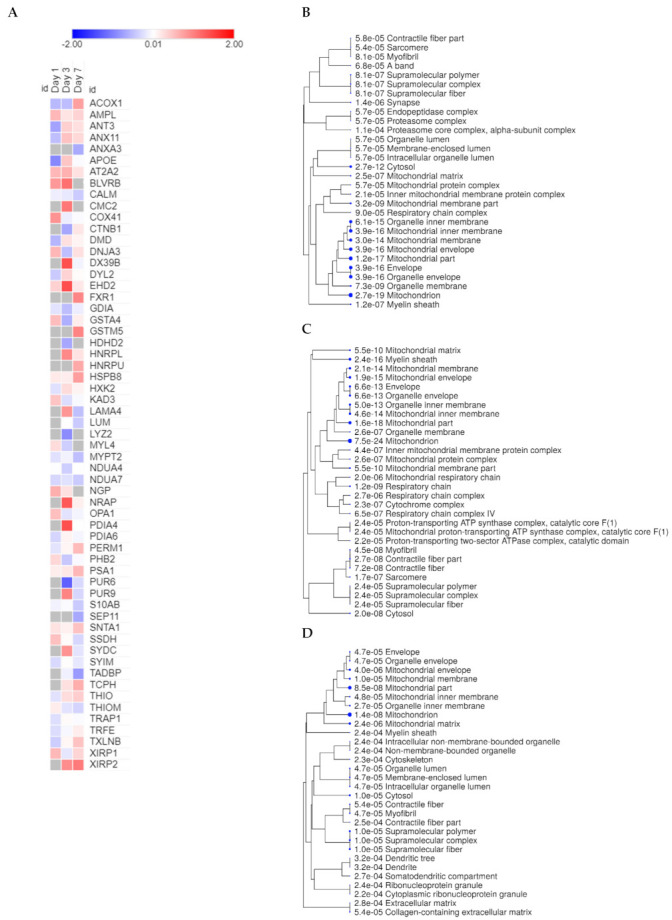
Heatmap of the top 20 significant DEPs and ShinyGO analysis of cellular components from the cross-comparative analysis. A cross-comparative analysis was done at each time point (1, 3 or 7 days after PCAL) between DMB vs. vehicle-treated mice to identify DEPs (*p* < 0.05). The log2 fold-change values from the top 20 DEPs at each time point were used to obtain the heatmap. A list containing all unique proteins and DEPs from each time point was uploaded on ShinyGO v0.61 and cellular components were obtained. Cellular components were clustered according to the amount of proteins they have in common and the size of the blue dots represent *p*-values (the bigger the size, the lower the *p*-value). (**A**) Heatmap of the top 20 DEPs at each time point; (**B**) top 30 cellular components of unique proteins and DEPs on day 1 post-PCAL; (**C**) top 30 cellular components of unique proteins and DEPs on day 3 post-PCAL; (**D**) top 30 cellular components of unique proteins and DEPs on day 7 post-PCAL.

**Figure 5 ijms-22-08711-f005:**
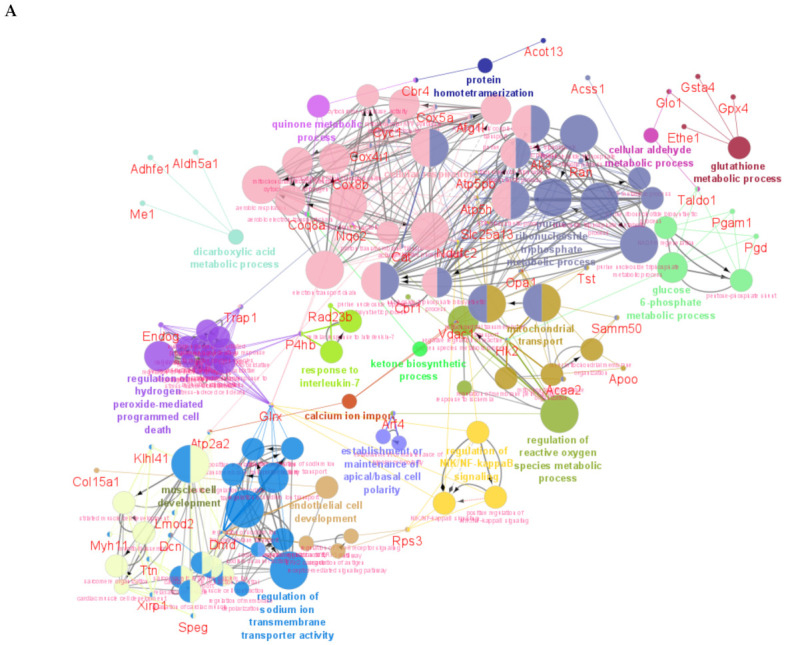
ClueGO analysis of biological processes of the unique proteins and DEPs between DMB vs. vehicle at each time point. A cross-comparative analysis was done at each time point (1, 3 or 7 days after PCAL) to identify the unique proteins and DEPs (*p* < 0.05). A list containing all unique proteins and DEPs for each time point was uploaded on ClueGO and biological processes were obtained. (**A**) Biological processes modulated by DMB on day 1 post-PCAL; (**B**) biological processes modulated by DMB on day 3 post-PCAL; (**C**) biological processes modulated by DMB on day 7 post-PCAL.

**Figure 6 ijms-22-08711-f006:**
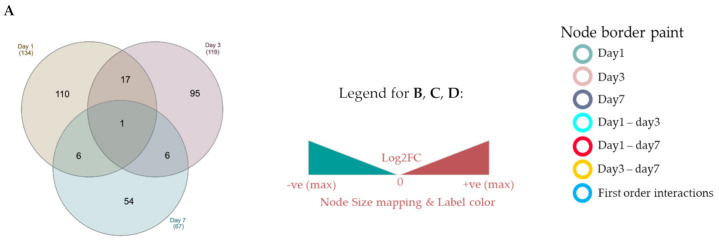
Protein–protein interaction (PPI) networks of unique proteins and DEPs identified in the cross-comparison analysis. A cross-comparative analysis was done at each time point (1, 3 or 7 days after PCAL) to identify unique proteins and DEPs (*p* < 0.05). (**A**) Venn diagram of unique proteins and DEPs identified in the DMB vs. vehicle comparison; (**B**) PPI network of unique proteins and DEPs on day 1 post-PCAL; (**C**) PPI network of unique proteins and DEPs on day 3 post-PCAL; (**D**) PPI network of unique proteins and DEPs on day 7 post-PCAL. * Protein unique to the DMB group for that day. # Protein unique to the vehicle group for that day.

**Table 1 ijms-22-08711-t001:** Unique canonical pathways with predicted *z*-scores throughout the course of remodeling.

	Vehicle		DMB	
	Pathway	*z*-Score	Pathway	*z*-Score
Day 1 vs. Naive	TCA Cycle II (Eukaryotic)	−2	Estrogen Receptor Signaling	1.342
	NAD signaling pathway	−1.633	GP6 Signaling Pathway	0.447
			Production of Nitric Oxide and Reactive Oxygen Species in Macrophages	0.447
			RhoA Signaling	1.633
			RhoGDI Signaling	−1.000
			Signaling by Rho Family GTPases	2.000
			Unfolded protein response	2.000
			Xenobiotic Metabolism AHR Signaling Pathway	−2.236
			Xenobiotic Metabolism CAR Signaling Pathway	0.378
			Xenobiotic Metabolism PXR Signaling Pathway	−1.633
Day 3 vs. Day 1	PFKFB4 Signaling Pathway	−1.000	Apelin Cardiomyocyte Signaling Pathway	−0.447
	RhoA signaling	2.000	Aspartate Degradation II	−2.000
	Calcium signaling	0.477	BAG2 Signaling Pathway	1.342
			Cardiac Hypertrophy Signaling	−0.447
			Cdc42 Signaling	−1.000
			Coagulation System	0.447
			Death Receptor Signaling	−0.447
			ERK/MAPK Signaling	2.000
			Estrogen Receptor Signaling	−2.309
			Gluconeogenesis I	−2.000
			Hepatic Fibrosis Signaling Pathway	−1.633
			HER-2 Signaling in Breast Cancer	2.000
			Insulin Secretion Signaling Pathway	−0.447
			Integrin Signaling	1.134
			NAD Signaling Pathway	−2.449
			Necroptosis Signaling Pathway	−2.000
			Nitric Oxide Signaling in the Cardiovascular System	1.000
			PI3K/AKT Signaling	1.000
			PPARα/RXRα Activation	−2.236
			Production of Nitric Oxide and Reactive Oxygen Species in Macrophages	1.000
			Regulation of Actin-based Motility by Rho	−1.000
			RhoGDI Signaling	1.342
			Role of PKR in Interferon Induction and Antiviral Response	−1.342
			Semaphorin Neuronal Repulsive Signaling Pathway	−2.236
			Senescence Pathway	−1.633
			Signaling by Rho Family GTPases	−0.447
			Superpathway of Methionine Degradation	−2.236
			TCA Cycle II (Eukaryotic)	−3.317
			Unfolded protein response	1.633
			Isoleucine Degradation I	−2.000
Day 7 vs. Day 3	Adrenomedullin signaling pathway	1.000	14-3-3-mediated Signaling	2.000
	Apelin Adipocyte Signaling Pathway	−2.000	Acetyl-CoA Biosynthesis I (Pyruvate Dehydrogenase Complex)	−2.000
	BAG2 Signaling Pathway	1.000	Complement System	0.816
	Breast Cancer Regulation by Stathmin1	−0.477	Gluconeogenesis I	1.342
	Cardiac Hypertrophy Signaling	1.000	Glutaryl-CoA Degradation	−2.236
	Cardiac Hypertrophy Signaling (Enhanced)	1.342	Glycolysis I	2.000
	Coagulation System	−1.342	Hepatic Fibrosis Signaling Pathway	1.342
	Coronavirus Pathogenesis Pathway	−2.000	ILK Signaling	1.732
	Death Receptor Signaling	2.000	Inhibition of ARE-Mediated mRNA Degradation Pathway	2.236
	Endocannabinoid Neuronal Synapse Pathway	1.000	Intrinsic Prothrombin Activation Pathway	−0.447
	Fcγ Receptor-mediated Phagocytosis in Macrophages and Monocytes	2.000	Isoleucine Degradation I	−2.236
	HIF1α Signaling	1.000	Ketogenesis	−1.342
	Leukocyte Extravasation Signaling	2.000	Ketolysis	−1.000
	Necroptosis Signaling Pathway	0.816	MSP-RON Signaling in Cancer Cells Pathway	2.236
	Opioid Signaling Pathway	1.000	Necroptosis Signaling Pathway	−0.447
	PPARα/RXRα Activation	−2.121	Senescence Pathway	−1.000
	RhoA Signaling	2.236	Tryptophan Degradation III (Eukaryotic)	−2.236
	RhoGDI Signaling	−2.000	Valine Degradation I	−2.646
	Role of NFAT in Cardiac Hypertrophy	2.000		
	Signaling by Rho Family GTPases	2.449		
	Unfolded protein response	2.000		
	Xenobiotic Metabolism CAR Signaling Pathway	1.000		
	Xenobiotic Metabolism PXR Signaling Pathway	−1.633		

**Table 2 ijms-22-08711-t002:** List of unique proteins identified in each group in the cross-comparative analysis.

Vehicle			DMB		
Day 1	Day 3	Day 7	Day 1	Day 3	Day 7
COFA1, COX8B, CTNB1, DCTN2, DUS3, EIF3B, EZRI, FAHD2, HNRPF, IMB1, KCC2D, KINH, LACE1, MARE2, MGDP1, MIC26, NDUC2, PDC6I, PSA7, RAN, RMD1, RS5, SAM50, SEPT2, STX3, TALDO, UBE2N, UBP5, 6PGD, APOC1, ARF4, C4BPA, CFAB, CHIL3, CMC2, CO4B, CRIP1, EGFR, FETUB, GPX41, HRG, ICAL, ITIH3, KLH41, LMOD2, LUM, MYH11, NEDD4, PRS10, PRS6A, RD23B, RS20, RS3, SYDC, TCPH, THTR, TOM1, UN45B, AN32A, CBR1, DHDH, FAHD1, GLRX1, HDHD2, HOT, NUCG, OXND1, PGS2, PIMT, PSA3, SIR5, SPEG, VP26A, CBR4, CLPP, DDX5, GPD1L, KV5AA, LGUL, MAOX, MIME, PYC, VPS29	CLPP, COX2, NP1L1, TSP1, CX7A2, XDH, FXR1, FBLI1	NUCG, GLRX3, AKAP9, CAND1, G3BP1, PRELP, VATG1, MYL4, MPPB	MLRA, MSRB2, PGFS, PSA6, NQO2, LYZ2	PDK4, LGMN, PMGE, RADI, CX6A1, COX8B, VATA, MTPN, RS3, TCPE, IC1, PSME2, PSMD1, PCBP2, SAM50, UN45B, DDX1, CPT1B, PLST, NAMPT, CAP2, IPYR, MECR, PFKAP, PSD13	ITIH3, CLPP, YBOX3, TAU, ARC1B, EIF3A, FBN1, NDUB1, PRS8, SYRC, PDK4

**Table 3 ijms-22-08711-t003:** Protein–protein interaction network parameters.

Day	Total Number of Nodes (Seed + Interacting Partners)	Number of Interactions	Unique Proteins and DEPs in Network (Seed Proteins)	High-Degree Proteins (Unique for That Day)
Day 1	198	612	43	Fancd2
				Atp2a2
				Eed
Day 3	768	3314	81	Fxr1
				Fancd2
				Actb
Day 7	922	4317	39	Mapt
				Actb
				Fancd2

## Data Availability

The mass spectrometry proteomics data have been deposited to the ProteomeXchange Consortium via the PRIDE partner repository with the dataset identifier PXD027867 and 10.6019/PXD027867.

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
