# Peer review of "Proteomics of Mouse Heart Ventricles Reveals Mitochondria and Metabolism as Major Targets of a Post-Infarction Short-Acting GLP1Ra-Therapy"

_ijms, 2021, doi:10.3390/ijms22168711_

Round 1

Reviewer 1 Report

The authors studied the role of DMB in post-infarction remodeling using proteomics combined with in silico analysis. The authors found that the mitochondria and oxidative metabolism are affected by DMB. While the proteomics provide comprehensive data, a further validation of the proteomics data should be conducted. This can be achieved by either measuring the protein levels of several mitochondrial proteins identified in the proteomics data  or measuring the mitochondrial function/activity in infarcted hearts treated with or without DMB.

Author Response

We thank you for providing us with your constructive criticisms regarding our manuscript submission. We have carefully reviewed each comment and provide our responses below the original comments, which were italicized.

Reviewer 1:

The authors studied the role of DMB in post-infarction remodeling using proteomics combined with in silico analysis. The authors found that the mitochondria and oxidative metabolism are affected by DMB. While the proteomics provide comprehensive data, a further validation of the proteomics data should be conducted. This can be achieved by either measuring the protein levels of several mitochondrial proteins identified in the proteomics data  or measuring the mitochondrial function/activity in infarcted hearts treated with or without DMB.

We thank the reviewer for the comments and for mentioning that our manuscript provides comprehensive data. We have previously demonstrated that mitochondrial turnover promoted by post-PCAL DMB therapy mitigated adverse cardiac remodeling in mice [1]. The induction of mitochondrial turnover was achieved by the enhancement of parkin-mediated mitophagy and mitochondrial biogenesis, represented by significant differences in the expression of mitochondrial proteins and genes, and by the increase of newly synthetized mitochondria. This way, we have previously proved that DMB modulates mitochondrial proteins and enhances the removal of dysfunctional mitochondria in the same in vivo model. We edited the “Introduction” section of this manuscript where we cite our previous publication for more clarification regarding the effects of DMB in mitochondria.

Reviewer 2 Report

de Freitas Germano and colleagues present the manuscript Proteomics of Mouse Heart Ventricles Reveals Mitochondria and Metabolism as Major Targets of a Post-Infarction Short-Acting GLP1Ra-Therapy. Sometimes, reviewers have the arduous task of providing feedback that takes a significant amount of time with a litany of corrections to improve the overall quality. I am happy to report that this is not one of those cases. It was a pleasure to review a manuscript. 

If I had to nitpick for corrections they are mostly in the results section and appear to be formatting, pdf conversion or loss of resolution in image generation

  • Figure 2 clarity is slightly lost on some lines, this may be an artifact of pdf conversion
  • Table S1 Columns "Vehicle" and "DMB" should have better spacing, column widths can be adjusted here for a clearer table. 
  •  Figure S2 same as figure 2 corrections, artifactually lost clarity
  • There is a floating letter "A" at the bottom of page 8 that belongs to figure 3 on the next page
  • Table 1 margins a partially bolded and un-bolded, probably another artifact from PDF conversion
  • Floating "A" again at the bottom of page 16, intended for figure 5 on page 17
  • Figure 5A and B node labelling are difficult to discern from overlapping text, some of the fonts are illegible in these images. This needs to be corrected
  • Floating "B" on page 17 as above
  • Figure S4A, the lower portion of text is missing on the pie chart labels
  •  
  • Materials and Methods, version of Cytoscape and ClueGo is missing

Author Response

We thank you for providing us with your constructive criticisms regarding our manuscript submission. We have carefully reviewed each comment and provide our responses below the original comments, which were italicized.

Reviewer 2:

de Freitas Germano and colleagues present the manuscript Proteomics of Mouse Heart Ventricles Reveals Mitochondria and Metabolism as Major Targets of a Post-Infarction Short-Acting GLP1Ra-Therapy. Sometimes, reviewers have the arduous task of providing feedback that takes a significant amount of time with a litany of corrections to improve the overall quality. I am happy to report that this is not one of those cases. It was a pleasure to review a manuscript.

We thank you for your kind comments and for the attention and time you committed to review our manuscript.

If I had to nitpick for corrections they are mostly in the results section and appear to be formatting, pdf conversion or loss of resolution in image generation

1) Figure 2 clarity is slightly lost on some lines, this may be an artifact of pdf conversion.

We apologize for that. We have replaced the figure with a clearer version. We believe it is an artifact of PDF conversion, as mentioned.

2) Table S1 Columns "Vehicle" and "DMB" should have better spacing, column widths can be adjusted here for a clearer table.

We thank the reviewer for pointing this out and have adjusted column widths.

3) Figure S2 same as figure 2 corrections, artifactually lost clarity.

We apologize for that. We have also replaced this figure with a clearer version.

4) There is a floating letter "A" at the bottom of page 8 that belongs to figure 3 on the next page.

We have corrected that.

5) Table 1 margins a partially bolded and un-bolded, probably another artifact from PDF conversion.

We believe this appears after PDF conversion since our Microsoft Word version does not show inconsistent table borders.

6) Floating "A" again at the bottom of page 16, intended for figure 5 on page 17.

We have corrected that.

7) Figure 5A and B node labelling are difficult to discern from overlapping text, some of the fonts are illegible in these images. This needs to be corrected.

We thank the reviewer for pointing this out. We understand networks can be confusing, but we believe they need to represent as much as possible the data obtained by proteomics. We have removed some biological processes from the total list of biological processes identified to obtain the networks we presented in the original submission. We opted to maintain our versions of the networks as they are, since we believe the biological processes that are displayed in Figures 5A-B complement each other, and it is hard to manage every single name displayed. We believe the pie-charts from supplementary figure S4, which are representative of the networks on figure 5 and include the complete list of biological processes identified on each day, can attenuate the complexity of the networks.

8) Floating "B" on page 17 as above.

We have corrected that.

9) Figure S4A, the lower portion of text is missing on the pie chart labels.

Thank you for pointing this out. We have corrected that and believe it is clearer now.

10) Materials and Methods, version of Cytoscape and ClueGo is missing.

We have included the version of Cytoscape and ClueGo in “Material and Methods”.

Reviewer 3 Report

In this study Dr. Juliana de Freitas Germano and her groups, combined a proteomics approach with in silico analysis to deeply understand the molecular basis of the early postinfarction remodeling in mouse, with and without administration of a GLP1R agonist, the DMB.

The experimental design was well organized and considered all the possible cross-evaluation between data collected from naive, vehicle-treated and DMB-treated mice, along different time points after PCAL surgery.

The bioinformatic analysis was performed using different softwares and and the obtained results were congruent each others, highlighting a strong involvement of mitochondria and mitochondria-associated pathways in the PCAL post-surgery recovery in mouse model. Only in few passages of the discussion it results difficult to follow the argumentation and the figures with only one reading, but this is due to the complexity of the images created by the softwares.

Nevertheless, I have some minor suggestions for the authors:

line 57: .."is" rat.. probably "in"

line 55-60: this part is not really clear, I suggest to rewrite it

line77: "10pmol/25g" is contradictory with Fig.1A were "10pmol/25mg" is reported

line 77: "(Reference)" does it mean that this value is considered as reference or it lacks of bibliographic reference?

Fig. 3: just a comment. For whom that are not familiar with IPA output, this representation can arise some doubts about the z-score values of vehicle and DMB DEPs. For example in Day 1 x Naive the Glycolysis I pathway z-score is -2 for Vehicle and -2 or -4 for DMB? If possible, I suggest to rearrange the image using 2 parallel lines for DMB and vehicle z-scores for each common DEP pathway.

line 410: graphic mistake on "°C"

line 420: mistake Af-ter

line 423-424: "For each sample, the volume corresponding to 150 μg of protein each sample 423 (150 μg) was mixed" the syntax of this sentence is not clear and some repetitions are present.

Author Response

We thank you for providing us with your constructive criticisms regarding our manuscript submission. We have carefully reviewed each comment and provide our responses below the original comments, which were italicized.

Reviewer 3:

In this study Dr. Juliana de Freitas Germano and her groups, combined a proteomics approach with in silico analysis to deeply understand the molecular basis of the early postinfarction remodeling in mouse, with and without administration of a GLP1R agonist, the DMB.

The experimental design was well organized and considered all the possible cross-evaluation between data collected from naive, vehicle-treated and DMB-treated mice, along different time points after PCAL surgery.

The bioinformatic analysis was performed using different softwares and and the obtained results were congruent each others, highlighting a strong involvement of mitochondria and mitochondria-associated pathways in the PCAL post-surgery recovery in mouse model. Only in few passages of the discussion it results difficult to follow the argumentation and the figures with only one reading, but this is due to the complexity of the images created by the softwares.

Nevertheless, I have some minor suggestions for the authors:

1) line 57: .."is" rat.. probably "in"

We have corrected that in the text.

2) line 55-60: this part is not really clear, I suggest to rewrite it

We have rewritten the text and hope it is clearer now.

3) line77: "10pmol/25g" is contradictory with Fig.1A were "10pmol/25mg" is reported

We thank the reviewer for pointing this out. The correct concentration is “10pmol/25g” as informed in the text. We have corrected Figure 1 accordingly.

4) line 77: "(Reference)" does it mean that this value is considered as reference or it lacks of bibliographic reference?

We have included the missing bibliographic reference in the text.

5) Fig. 3: just a comment. For whom that are not familiar with IPA output, this representation can arise some doubts about the z-score values of vehicle and DMB DEPs. For example in Day 1 x Naive the Glycolysis I pathway z-score is -2 for Vehicle and -2 or -4 for DMB? If possible, I suggest to rearrange the image using 2 parallel lines for DMB and vehicle z-scores for each common DEP pathway. Redo graphs

We agree with the reviewer and have modified Figure 3 accordingly. We believe it is clearer now and more representative of the results.

6) line 410: graphic mistake on "°C"

We have corrected that.

7) line 420: mistake Af-ter

We have fixed it accordingly.

8) line 423-424: "For each sample, the volume corresponding to 150 μg of protein each sample 423 (150 μg) was mixed" the syntax of this sentence is not clear and some repetitions are present.

We agree the sentence was not clear and have corrected that.

We thank the reviewer for the positive feedback and for pointing out minor suggestions that made our manuscript clearer and more complete.

Round 2

Reviewer 1 Report

The authors have addressed my comments.